# Dig a Hole and Fill in Sand: Adversary and Hiding Decoupled Steganography

Weixuan Tang
Institute of Artificial Intelligence,
Guangzhou University
Guangzhou, China
tweix@gzhu.edu.cn

Haoyu Yang
Institute of Artificial Intelligence,
Guangzhou University
Guangzhou, China
haoyuyang@e.gzhu.edu.cn

Yuan Rao
Institute of Artificial Intelligence,
Guangzhou University
Guangzhou, China
raoyuan@gzhu.edu.cn

Zhili Zhou*
Institute of Artificial Intelligence,
Guangzhou University
Guangzhou, China
zhouzhili@163.com

Fei Peng
Institute of Artificial Intelligence,
Guangzhou University
Guangzhou, China
eepengf@gmail.com

## Abstract

Deep steganography is a technique that imperceptibly hides secret information into image by neural networks. Existing networks consist of two components, including a hiding component for information hiding and an adversary component for countering against steganalyzers. However, these two components are two ends of the seesaw, and it is difficult to balance the tradeoff between message extraction accuracy and security performance by joint optimization. To address the issues, this paper proposes a steganographic method called AHDeS (Adversary-Hiding-Decoupled Steganography) under the Dig-and-Fill paradigm, wherein the adversary and hiding components can be decoupled into an optimization-based adversary module in the digging process and an INN-based hiding network in the filling process. Specifically in the training stage, the INN is first trained for acquiring the ability of message embedding. In the deployment stage, given the well-trained and fixed INN, the cover image is first iteratively optimized for enhancing the security performance against steganalyzers, followed by the actual message embedding by the INN. Owing to the reversibility of the INN, security performance can be enhanced without sacrificing message extraction accuracy. Experimental results show that AHDeS can achieve the state-of-the-art security performance and visual quality while maintaining satisfied message extraction accuracy.

## CCS Concepts

• **Information systems** → **Multimedia information systems**; • **Security and privacy** → **Security services**.

## Keywords

Multimedia steganography; Information hiding; Invertible neural networks

**ACM Reference Format:**
Weixuan Tang, Haoyu Yang, Yuan Rao, Zhili Zhou, and Fei Peng. 2024. Dig a Hole and Fill in Sand: Adversary and Hiding Decoupled Steganography. In *Proceedings of the 32nd ACM International Conference on Multimedia (MM '24), October 28-November 1, 2024, Melbourne, VIC, Australia.* ACM, New York, NY, USA, 9 pages. https://doi.org/10.1145/3664647.3681330

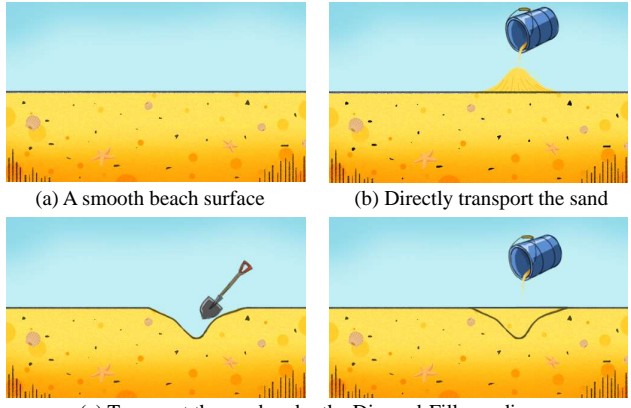

(a) A smooth beach surface      (b) Directly transport the sand

(c) Transport the sand under the Dig-and-Fill paradigm

**Figure 1: Illustration of the Dig-and-Fill Paradigm.**

*Corresponding author.

## 1 Introduction

Multimedia steganography is the art and science of concealing secret messages within multimedia carriers, where digital images are the commonly used carriers. Traditional steganographic methods were designed under the distortion minimization framework [10], which can be formulated as minimizing distortion function under payload constraint. In the past decades, different distortion functions have been proposed, which were based on heuristic principles [15, 19] or statistical models [11, 13]. To further improve the security performance, deep learning techniques have been introduced, and the distortion functions could be automatically learned by GAN

(Generative Adversarial Networks) [27, 33], RL (Reinforcement Learning) [23, 24], or AE (Adversarial Examples) [2, 26]. Practical steganographic codes, such as STC (Syndrome Trellis Code) [9], can be applied for message embedding according to the defined distortion functions.

However, due to the constraint of distortion minimization framework, the hiding capacities of the above steganographic methods were relatively low. To obtain higher capacities, deep steganography has been proposed, which utilized neural networks to implement message embedding and extraction, instead of practical steganographic codes. These neural networks consist of a hiding component for message embedding and an adversary component for countering against steganalyzers. Baluja [1] proposed the first deep image hiding method. Hayes *et al.* [14] introduced adversarial training into deep steganography. Zhu *et al.* [37] introduced noise layer into HiDDeN for robust steganography. Zhang *et al.* [36] proposed SteganoGAN with residual and dense architecture. Considering that the reversibility of INN (Invertible Neural Networks) is suitable for steganography task, Jing *et al.* [16] and Lu *et al.* [21] respectively proposed steganographic method based on INN, which implemented message hiding and extraction by the forward calculation and reverse calculation of INN. Lan *et al.* [18] utilized INN to hide secret information in the DCT domain to resist JPEG compression. In the above deep steganographic methods, the hiding component and adversary component are jointly trained. However, higher payload information hiding would lead to more modification artifacts, which can increase the risk of being detected by steganalyzers. Therefore, these two components are two ends of the seesaw, and it is difficult to balance the tradeoff between message extraction accuracy and security performance by joint optimization with multiple loss functions.

To address the above limitations, this paper first proposes the Dig-and-Fill paradigm, as shown in Fig. 1. The key idea is inspired from the case of imperceptible sand transportation. If the porter has some prior knowledge of the sand to be transported, such as the sand's volume, then the porter can dig a hole with equal volume in advance, and then subsequently fill the hole with sand. By this manner, the beach surface can still be kept unchanged, and the transportation behaviour can be well hidden. In the case of steganography, the adversary and hiding components can be decoupled into the digging and filling processes. Specifically, if the steganographer is aware of the hiding pattern in the filling process, then the steganographer can optimize the cover image in advance in the digging process, so that the optimized cover image embedded with secret messages can obtain satisfied security performance.

Under such paradigm, a steganographic method called AHDeS (Adversary-Hiding-Decoupled Steganography) is proposed. AHDeS is an optimization-based and model-based hybrid method, wherein the adversary component and hiding component are implemented as an optimization-based adversary module and an INN-based hiding network, respectively. Specifically, the INN is first trained for obtaining the ability of message embedding. Afterwards, given the well-trained and fixed INN, the cover image is iteratively optimized for enhancing the security performance against steganalyzers in the filling process, followed by actual information hiding by INN in the digging process. As long as the INN is fixed, owing to the reversibility and the dual branch structure of INN, security performance can

be enhanced without sacrificing message extraction accuracy. The contributions of this work are summarized as follows:

- A Dig-and-Fill paradigm is designed, wherein the adversary and hiding components of steganographic methods can be decoupled into the digging and filling processes and independently optimized.
- A steganographic method called AHDeS is proposed, which can take advantage of the reversibility and the dual branch structure of INN, and iteratively optimize the cover image for enhancing the security performance without sacrificing message extraction accuracy.
- A frequency compensation mechanism is deigned, wherein the optimization-based adversary module strives to preserve the high-frequency components neglected by the INN-based hiding network.
- Experimental results show that AHDeS can significantly improve the security performance while maintaining satisfied message extraction accuracy.

## 2 Related work

### 2.1 Image Steganography

With the rapid development of deep learning technique, image steganographic methods based on deep learning have received great attention. Baluja [1] proposed the first deep image hiding method, wherein the secret image and cover image were concatenated and fed into the encoder to generate the stego image, and then stego image was fed into the decoder to reveal the secret image. Hayes *et al.* [14] introduced adversarial training into deep steganography, wherein the generator was utilized to generate stego image, and the discriminator was utilized to distinguish between cover and stego images. Zhu *et al.* [37] proposed HiDDeN, wherein the noise layer was introduced in the training stage to enhance the robustness of steganographic method. Zhang *et al.* [36] proposed SteganoGAN, wherein the residual and dense architecture was applied to improve the payload of steganographic method. Yu [34] proposed ABDH, wherein the attention mechanism was applied to find the inconspicuous areas of cover images. Zhang *et al.* [35] proposed UDH, wherein the secret image was projected as universal adversarial perturbation, and was embedded into the cover image in a cover-agnostic manner. Tan *et al.* [25] proposed CHAT-GAN, where the channel attention mechanism was applied to improve the quality of stego image and the error-correcting algorithm was applied to improve the message extraction accuracy.

The above methods are modification-based methods. Another type are generative methods, which generate stego according to latent noise by generative models, such as [5]. It maps secret message into latent noise without changing its distribution, and thus the distribution of stego and cover image (generated image without secret message) could be the same. Despite this, image quality for cover and stego images depends on the generative models, and poor image quality would raise suspicion. This paper focuses on modification-based methods.

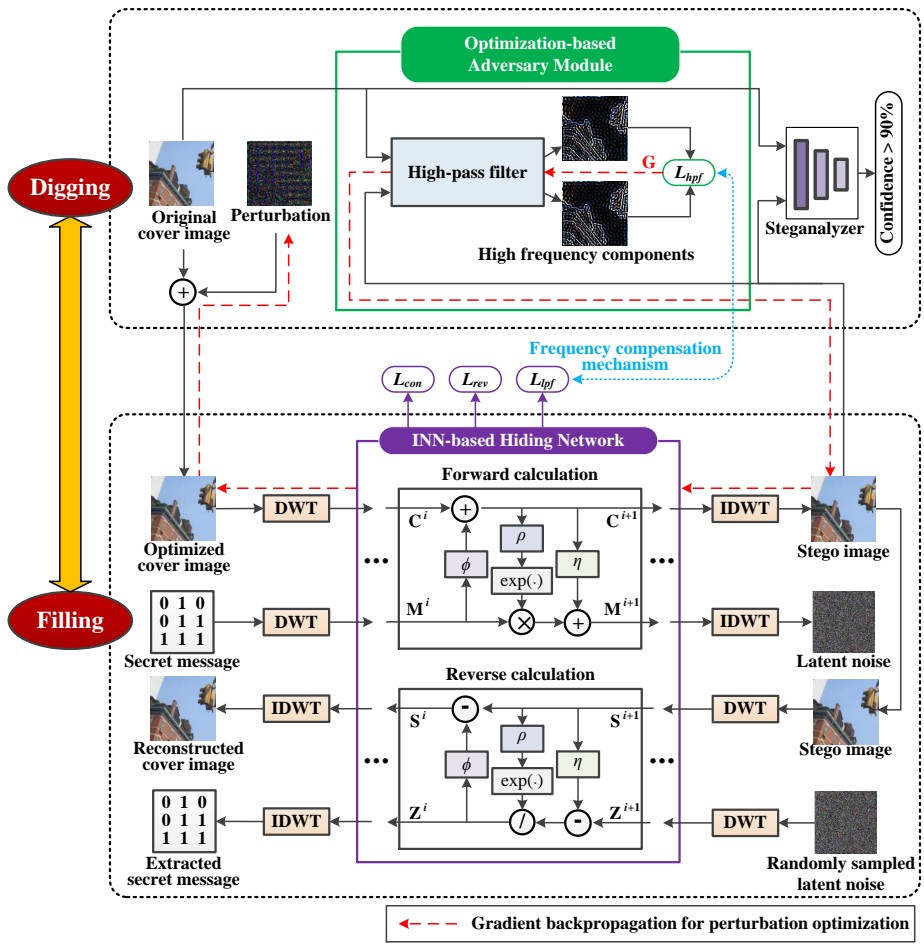

Figure 2: Illustration of the proposed AHDeS.

## 2.2 Invertible Neural Network

Dinh *et al.* proposed the concept of INN [7], and designed the coupling layers to improve its representation ability [8]. Kingma *et al.* [17] proposed Glow, which utilized invertible $1 \times 1$ convolution to reverse the ordering of channels. Ouderaa *et al.* [28] proposed INN with low memory complexity in image-to-image translation task. Wang *et al.* [29] applied INN in digital image compression task, which reduced information loss. Xiao *et al.* [31] proposed to model the rescaling process as invertible transformation between high-resolution and low-resolution images. Ma *et al.* [22] incorporated INN into blind watermarking to learn the joint representation between watermark embedding and extraction.

In recent years, INN has also been applied in the field of image steganography. Jing *et al.* [16] proposed HiNet, which implemented message hiding and extraction by the forward calculation and reverse calculation of INN. Lu *et al.* [21] also applied INN to the field of steganography. Guan *et al.* [12] proposed DeepMIH, which can sequentially hide multiple secret images under the guidance of importance map module. Lan *et al.* [18] proposed to hide secret

information in the DCT domain of cover image and designed a mutual information loss to constrain the flow of information.

## 3 Dig-and-Fill Paradigm

Existing deep steganographic methods consist of two essential components, including an adversary component for countering against steganalyzers and a hiding component for message embedding. In previous methods, the adversary and hiding components are jointly trained, which increase the optimization difficulties. To overcome the above limitation, the Dig-and-Fill paradigm is proposed, which can decouple these two components for independent optimization.

Suppose that the task is to transport a specific volume of sand from the bucket to the beach while avoiding leaving obvious transportation traces on the beach. If the porter directly transports the sand, the beach surface would be severely changed and arouse suspicion, as shown in Fig. 1 (b). However, if the porter has some prior knowledge of the sand to be transported, it is possible to transport the sand in an imperceptible manner. For example, if the porter knows the sand's volume, then the porter can dig a hole with equal volume in advance. Afterwards, the sand can be transported into

such hole, and then the beach surface can be kept unchanged, as shown in Fig. 1 (c). The key idea of such Dig-and-Fill paradigm is that if the filling process can be pre-determined, then the digging process can be settled according to the filling process, and the transportation behavior can be well hidden.

In the field of image steganography, the task is to hide secret information into the cover image while maintaining the stego image undetected by steganalyzer. Under the Dig-and-Fill paradigm, the steganographer can first optimize the cover image in the digging process and then embed secret messages into the optimized cover image in the filling process. If the steganographer is fully aware of the hiding pattern in the filling process, then the steganographer can optimize the cover image in advance in the digging process, so that the optimized cover image embedded with secret messages can obtain satisfied security performance. By this means, the adversary and hiding components can be effectively decoupled into the digging and filling processes. The remaining challenge is to design a network via which can optimize the digging process while keeping the effect of filling process unchanged.

## 4 Proposed Method

### 4.1 Method Overview

Following the Dig-and-Fill paradigm, a steganographic method called AHDeS (Adversary-Hiding-Decoupled Steganography) is proposed, as shown in Fig. 2. In AHDeS, the INN-based hiding network is first trained to acquiring the ability of message embedding, and then based on the well trained INN, the optimization-based adversary module is utilized to optimize the cover image for enhancing security performance in the digging process, followed by the actual message embedding by INN the filling process.

Specifically, the INN is first trained to obtaining the ability of information hiding and extraction via forward calculation and reverse calculation, respectively. Such an INN is of dual-branch structure. In the forward calculation, it takes in the cover image $\mathbf{C}$ and the secret message $\mathbf{M}$, and outputs the stego image $\mathbf{S}$ and the latent noise $\mathbf{Z}$. In the reverse calculation, it takes in $\mathbf{S}$ and the randomly sampled latent noise $\widetilde{\mathbf{Z}}$, and outputs the reconstructed cover image $\widetilde{\mathbf{C}}$ and the extracted secret message $\widetilde{\mathbf{M}}$. Benefiting from the reversible properties of INN, $\widetilde{\mathbf{M}}$ is of great message extraction accuracy. Given the well-trained and fixed INN, the adversary module iteratively optimizes the perturbation $\mathbf{P}$, which is added on the original cover image $\mathbf{C}^{ori}$ to obtain the optimized cover image $\mathbf{C}^{opt}$. Such $\mathbf{C}^{opt}$ is further fed into the fixed INN to obtain the stego image $\mathbf{S}$. The perturbation $\mathbf{P}$ is iteratively optimized according to the frequency compensation mechanism, which tries to preserve the high-frequency components neglected by the fixed INN. The iteration ends until the stego image $\mathbf{S}$ can be classified as cover image with high confidence by a pre-trained CNN-based steganalyzer. Owing to the dual-branch structure and reversible properties of INN, perturbing the original cover image $\mathbf{C}^{ori}$ can contribute to generating stego image $\mathbf{S}$ with higher anti-steganalysis ability in the forward calculation, and barely affect extracting secret messages $\widetilde{\mathbf{M}}$ extremely close to $\mathbf{M}$ in the reverse calculation. Therefore, the optimization-based adversary module can enhance security performance against steganalyzers without sacrificing message extraction accuracy.

## 4.2 INN-based Hiding Network

The reversibility of INN is suitable for steganography task. It takes in the cover image $\mathbf{C}$ and the secret message $\mathbf{M}$ and outputs the stego image $\mathbf{S}$ and the latent noise $\mathbf{Z}$ as

$$\mathbf{S}, \mathbf{Z} = h_\theta (\mathbf{C}, \mathbf{M}) . \tag{1}$$

In the reverse calculation, it takes in $\mathbf{S}$ and the randomly sampled latent noise $\widetilde{\mathbf{Z}}$, and outputs the reconstructed cover image $\widetilde{\mathbf{C}}$ and extracted secret message $\widetilde{\mathbf{M}}$ as

$$\widetilde{\mathbf{C}}, \widetilde{\mathbf{M}} = h_\theta^{-1} \left( \mathbf{S}, \widetilde{\mathbf{Z}} \right) . \tag{2}$$

Such an INN consists of 16 invertible blocks. Specifically, the $k$-th block's affine transformation in forward calculation can be formulated as

$$\mathbf{C}^{(k+1)} = \mathbf{C}^{(k)} + \phi \left( \mathbf{M}^{(k)} \right) , \tag{3}$$

$$\mathbf{M}^{(k+1)} = \mathbf{M}^{(k)} \odot \exp \left( \lambda \cdot \rho \left( \mathbf{C}^{(k+1)} \right) \right) + \eta \left( \mathbf{C}^{(k+1)} \right) , \tag{4}$$

where $\mathbf{C}^{(1)} = \mathbf{C}$, $\mathbf{M}^{(1)} = \mathbf{M}$, $\mathbf{C}^{(17)} = \mathbf{S}$, $\mathbf{M}^{(17)} = \mathbf{Z}$. And the $k$-th block's affine transformation in reverse calculation can be formulated as

$$\widetilde{\mathbf{Z}}^{(\mathbf{k})} = \left( \widetilde{\mathbf{Z}}^{(k+1)} - \eta \left( \mathbf{S}^{(k+1)} \right) \right) \oslash \exp \left( \lambda \cdot \rho \left( \mathbf{S}^{(k+1)} \right) \right) , \tag{5}$$

$$\mathbf{S}^{(k)} = \mathbf{S}^{(k+1)} - \phi \left( \widetilde{\mathbf{Z}}^{(\mathbf{k})} \right) , \tag{6}$$

where $\mathbf{S}^{(17)} = \mathbf{S}$, $\widetilde{\mathbf{Z}}^{(17)} = \widetilde{\mathbf{Z}}$, $\mathbf{S}^{(1)} = \widetilde{\mathbf{C}}$, $\widetilde{\mathbf{Z}}^{(1)} = \widetilde{\mathbf{M}}$. Note that $\phi$, $\rho$, and $\eta$ denote 3 CNNs with same structure but different parameters. Cover image is denoted as $\mathbf{C} \in \{0, ..., 255\}^{H,W,N}$. Binary bitstream is converted into three-dimensional format as $\mathbf{M} \in \{0, 1\}^{H,W,N'}$. As for 1.0 bpp (bit per pixel), $N$ is equal to $N'$. $\mathbf{C}$ and $\mathbf{M}$ are processed by DWT (Discrete Wavelet Transform) into size $(H/2, W/2, 4N)$, and further fed into the INN.

The loss function of INN is formulated as

$$L_{\text{INN}} = \lambda_c L_{con} + \lambda_r L_{rev} + \lambda_l L_{lpf}. \tag{7}$$

$L_{con}$ is the concealing loss that makes the stego image similar to the original cover image, and is formulated as

$$L_{con} = \frac{1}{H \times W \times N} ||\mathbf{C}^{ori} - \mathbf{S}||^2. \tag{8}$$

$L_{rev}$ is the revealing loss for successful message extraction, and is formulated as

$$L_{rev} = \frac{1}{H \times W \times N} ||\mathbf{M} - \widetilde{\mathbf{M}}||^2. \tag{9}$$

$L_{lpf}$ is the low-pass-filter loss that tries to preserve the low frequency sub-band unchanged, and is formulated as

$$L_{lpf} = \frac{1}{H/2 \times W/2 \times N} ||\mathbf{C}_{LL} - \mathbf{S}_{LL}||^2, \tag{10}$$

where $\mathbf{C}_{LL}$ and $\mathbf{S}_{LL}$ denote the low frequency sub-band after wavelet decomposition for $\mathbf{C}^{ori}$ and $\mathbf{S}$, respectively.

## 4.3 Adversary Module

Benefitting from the dual branch structure and the reversibility of INN, from Eq. (1) and Eq. (2), it can be observed that given a fixed INN and in the ideal situation that $\widetilde{\mathbf{Z}}$ and $\mathbf{Z}$ are the same, no matter what perturbation is added on the cover image, the extracted secret message $\widetilde{\mathbf{M}}$ and the original secret message $\mathbf{M}$ would still be the same. Inspired by such property, the optimization-based adversary module is designed to enhance the anti-steganalysis ability without sacrificing message extraction accuracy.

The key idea of the adversary module is that given a well-trained and fixed INN, the perturbation $\mathbf{P}$ is optimized and then added on the original cover image $\mathbf{C}^{ori}$, so that the optimized cover image embedded with secret messages can obtain better security performance against steganalyzer. Note that according to $L_{lpf}$ in Eq. (10), the INN strives to preserve the low frequency components unchanged, and tends to modify the high frequency components for information hiding. Although high frequency components are more suitable to hide secret information than low frequency components, inappropriate and extensive modifications would still lead to poor anti-steganalysis performance. Therefore, a frequency compensation mechanism is deigned in the adversary module, which strives to preserve the high frequency components neglected by the INN-based hiding network. Specifically, given the well-trained and fixed INN, the adversary module aims to preserve the high frequency components between the original cover image $\mathbf{C}^{ori}$ and the stego image $\mathbf{S}$, and the loss function is formulated as

$$L_{hpf} = \frac{1}{H \times W \times N} ||\mathcal{F}(\mathbf{C}^{ori}) - \mathcal{F}(\mathbf{S})||^2, \quad (11)$$

where $\mathcal{F}$ denotes the operation of obtaining the high frequency components via DFT (Discrete Fourier Transform).

Note that the perturbation $\mathbf{P}$ is added on the original cover image $\mathbf{C}^{ori}$ to obtain the optimized cover image $\mathbf{C}^{opt}$, which is further fed into the INN-based hiding network to obtain the stego image $\mathbf{S}$. Such $\mathbf{P}$ is iteratively optimized by means of minimizing Eq. (11). Specifically, in each iteration, the gradients $\mathbf{G}$ of the loss function $L_{hpf}$ with respect to $\mathbf{P}$ is calculated, and $\mathbf{P}$ is updated by subtracting $\mathbf{G}$. The iterations are terminated until the stego image can obtain satisfied security performance. To evaluate the security performance, a CNN-based steganalyzer is first trained and then fixed. The termination condition is that the stego image is judged as cover image with confidence higher than a specific threshold or iteration number exceeds a certain value.

## 4.4 Training and Deployment Strategy

The proposed AHDeS is a model-based and optimization-based hybrid method. The training strategy is given in Algorithm 1. In general, in the training stage, the INN is trained to obtain the ability of message embedding and extraction. The deployment strategy is given in Algorithm 2. In the deployment stage, the well-trained INN is fixed, and perturbation optimization for cover image in digging process occurs before message embedding in filling process. Therefore, the only network that needs to be trained is the INN. To sufficiently train the INN in AHDeS, a perturbation adaptive training strategy is proposed. Such strategy contains two phases. The first phase aims to make the INN acquire the basic function of message embedding. It takes the original cover image $\mathbf{C}^{ori}$ as input,

and its parameters are updated by minimizing Eq. (7). Afterwards, the second phase aims to adapt the INN to the Dig-and-Fill paradigm, and let the INN learn to process the optimized cover image $\mathbf{C}^{opt}$. Specifically, in each training iteration, the adversary module is first applied to iteratively add $\mathbf{P}$ on $\mathbf{C}^{ori}$. And then the INN takes $\mathbf{C}^{opt}$ as input, and its parameters are updated by minimizing Eq. (7). By this means, the well-trained and fixed INN is adaptive to the Dig-and-Fill paradigm and can be regarded as the prior knowledge for the adversary module.

---

**Algorithm 1** Training Strategy

---

1: **Require:** CNN-based steganalyzer $D$, confidence threshold $Q$, iteration number $N_1$, $N_2$;
2: **Input:** Original cover image $\mathbf{C}^{ori}$, secret message $\mathbf{M}$;
3: **Output:** INN with parameters $\theta$;
4: Initialize $\theta$;
5: **for** $n = 0$ to $N_1$ **do**       // First phase.
6:     $(\mathbf{S}, \mathbf{Z}) \leftarrow h_\theta(\mathbf{C}^{ori}, \mathbf{M})$       // Forward calculation.
7:     $(\widetilde{\mathbf{C}}, \widetilde{\mathbf{M}}) \leftarrow h_\theta^{-1}(\mathbf{S}, \widetilde{\mathbf{Z}})$       // Reverse calculation.
8:     Update $\theta$ with loss function $L_{\text{INN}}$;
9: Initialize the perturbation $\mathbf{P}$;
10: **for** $n = 0$ to $N_2$ **do**       // Second phase.
11:     $\mathbf{C}^{opt} = \mathbf{C}^{ori} + \mathbf{P}$
12:     $(\mathbf{S}, \mathbf{Z}) \leftarrow h_\theta(\mathbf{C}^{opt}, \mathbf{M})$       // Forward calculation.
13:     $(\widetilde{\mathbf{C}}, \widetilde{\mathbf{M}}) \leftarrow h_\theta^{-1}(\mathbf{S}, \widetilde{\mathbf{Z}})$       // Reverse calculation.
14:     $q \leftarrow D(\mathbf{S})$ //Obtain confidence of judging $\mathbf{S}$ as $\mathbf{C}^{ori}$.
15:     **if** $q \geq Q$ or $n == N_2$ **then**
16:         **break**
17:     Update $\theta$ with loss function $L_{\text{INN}}$;
18:     Update $\mathbf{P}$ with loss function $L_{hpf}$;

---

**Algorithm 2** Deployment Strategy

---

1: **Require:** CNN-based steganalyzer $D$, confidence threshold $Q$, iteration number $N$, well-trained INN with parameters $\theta$;
2: **Input:** Original cover image $\mathbf{C}^{ori}$, secret message $\mathbf{M}$;
3: **Output:** Stego image $\mathbf{S}$;
4: Initialize the perturbation $\mathbf{P}$;
5: **for** $n = 0$ to $N$ **do**
6:     $\mathbf{C}^{opt} = \mathbf{C}^{ori} + \mathbf{P}$       // Digging process.
7:     $(\mathbf{S}, \mathbf{Z}) \leftarrow h_\theta(\mathbf{C}^{opt}, \mathbf{M})$       // Filling process.
8:     $(\widetilde{\mathbf{C}}, \widetilde{\mathbf{M}}) \leftarrow h_\theta^{-1}(\mathbf{S}, \widetilde{\mathbf{Z}})$
9:     $q \leftarrow D(\mathbf{S})$ //Obtain confidence of judging $\mathbf{S}$ as $\mathbf{C}^{ori}$.
10:     **if** $q \geq Q$ or $n == N$ **then**
11:         **break**
12:     Update $\mathbf{P}$ with loss function $L_{hpf}$;

---

## 5 Experiments

## 5.1 Experimental Setup

**Dataset.** The COCO dataset [20] is utilized in the experiments. Specifically, 5,000 images are utilized to train the deep-learning-based steganographic models in the training stage and 10,000 images are utilized to generate stego images in the deployment stage.

These 10,000 images are split into training, validation, and testing set for security performance evaluation by steganalyzers according to the proportion of 5:1:4. As for the steganalyzer applied in the adversary module, 10,000 images are utilized to train such steganalyzer. All images are central cropped into $256 \times 256$.

Implementation details. The Adam optimizer with default settings is utilized to to optimize the proposed AHDeS. For the INN-based hiding network, the initial learning rate is set to $1 \times 10^{-5.2}$, and the batch size is set to 16. For the optimization-based adversary module, the learning rate is set to $1 \times 10^{-4}$, and the batch size is set to 1. $\lambda_c$, $\lambda_r$, and $\lambda_f$ in Eq. (7) are all set to 1.0. The first and second phases of training process of INN consist of $1,000$ and 200 epochs, respectively. In the termination condition of perturbation optimization, the threshold for steganalyzer's confidence and iteration number is set to 90% and 200, respectively. Note that we simulate real-world scenarios of secret information transmission, wherein digital images in integer type are applied as carrier. In the deep-learning-based steganographic methods, the neural networks receives original images in integer type and outputs temporal images in floating type, which are further rounded into stego image in integer type. This rounding process leads to a certain degree of information loss and may further result in performance degradation of secret message extraction accuracy. Such stego images are evaluated from different aspects, including security performance, decoding performance, and visual quality. The experiments are implemented by PyTorch and executed on Tesla V100 GPU card.

Steganographic methods and steganalyzers. Five steganographic methods are compared, including HiDDeN [37], SteganoGAN [36], CHAT-GAN [25], HiNet [16], and the proposed AHDeS. Noted that network with specific structure can hide both bitstream and image. For example, HiNet is originally used to hide image, while [18], which has similar structure as HiNet, is used to hide bitstream. Therefore, as long as the hidden content is the same, the comparison would be fair. Five steganalyzers are utilized to evaluate the security performance of steganographic methods, including XuNet [32], SRNet [4], LWENet [30], CovNet [6], and StegExpose [3].

## 5.2 Performance Evaluation

In this part, the performance of steganographic methods are evaluated from two aspects, including security performance and decoding performance.

Four steganalyzers are adopted to evaluate the security performance of steganographic methods. Specifically, CovNet is utilized to evaluate the security performance of the generated stego images in the adversary module in the proposed AHDeS. Therefore, from the perspective of steganographer of AHDeS, CovNet is regarded as the known steganalyzer, while XuNet, SRNet, and LWENet are regarded as the unknown steganalyzers. The experimental results are shown in Table 1. It can be observed that the proposed AHDeS achieves the best security performance against both known and unknown steganalyzers in all cases. Compared with HiNet, AHDeS can obtain improvement of 16.80%, 5.57%, 6.62%, and 6.83% against XuNet, SRNet, LWENet, and CovNet, respectively. The traditional handcrafted steganalyzer is also adopt for performance evaluation. AUC is 0.509 for AHDeS, and 0.592, 0.567, 0.581, 0.573 for the rest. Noted that lower AUC indicates better security performance.

**Table 1: Detection error rate of steganographic methods and bit error rate between the original message and the extracted message (%).**

| Methods | CovNet | XuNet | SRNet | LWENet | BER |
|---|---|---|---|---|---|
| HiDDeN | 0.25 | 0.53 | 0.24 | 0.03 | 40.70 |
| SteganoGAN | 0.32 | 1.07 | 0.44 | 0.07 | 1.47 |
| CHAT-GAN | 0.30 | 0.89 | 0.35 | 0.03 | 1.18 |
| HiNet | 2.16 | 17.05 | 7.53 | 0.75 | **0.26** |
| **AHDeS** | **8.99** | **33.85** | **13.10** | **7.37** | 0.85 |

**Table 2: Objective evaluation of visual quality for steganographic methods.**

| Methods | PSNR | SSIM |
|---|---|---|
| HiDDeN | 37.29 | 0.9822 |
| SteganoGAN | 42.61 | 0.9904 |
| CHAT-GAN | 43.34 | 0.9909 |
| HiNet | 42.95 | 0.9893 |
| **AHDeS** | **45.07** | **0.9938** |

BER (Bit Error Rate) between the original message and the extracted message is utilized to evaluate the decoding performance of steganographic methods. The results are shown in Table 1. It can be observed that the BER of HiNet and AHDeS is close to 0 and is superior to the other methods, indicating that the proposed AHDeS can transmit secret messages in a nearly lossless manner.

We also analyze the relationship between payload, security performance, and decoding performance. Specifically, in the case of 1 bpp, the detection error rate of CovNet is 8.99% and the BER is 0.85%. While in the case of 2 bpp, the detection error rate is 7.27% and the BER is 0.95%. It can be observed that the higher the payload, the lower the security performance and the decoding performance. Noted that although INN has the inverse property, its BER is affected by the rounding operation for the forward calculation's output. To hide larger payload of messages, a cover image's pixel has to carry more bits in average, and thus given a specific modification range for such pixel, the modification range for each bit would be smaller and thus could be more easily erased by the rounding operation. Besides, higher payload would introduce more artifacts and thus decrease security performance.

## 5.3 Visual Quality Evaluation

In this part, the visual quality of the stego image is evaluated from the objective evaluation and the subjective evaluation.

For subjective evaluation, we visualize the difference between the cover and stego images. The cover image in AHDeS is the original cover image. The results are given in Fig. 3. It can be observed that the difference for proposed AHDeS are nearly invisible. Such phenomenon indicates that AHDeS can generate stego images in an imperceptible manner.

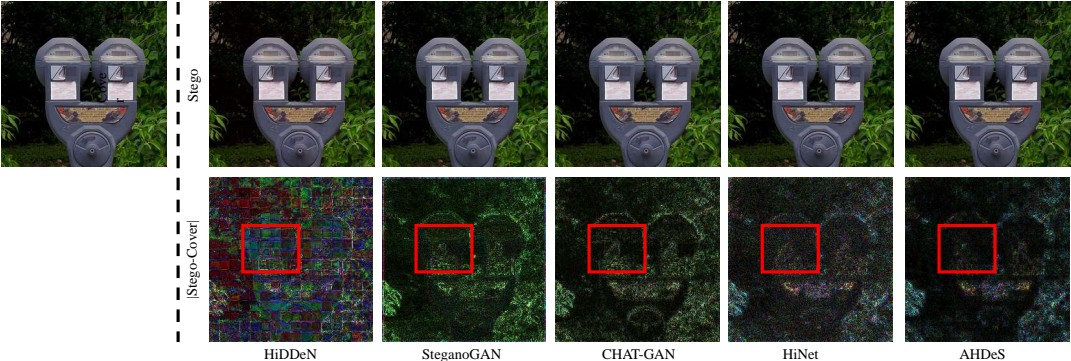

**Figure 3: Subjective evaluation of visual quality for steganographic methods. The difference is magnified by 20 times.**

For objective evaluation, we evaluate the visual quality by PSNR (peak-signal-to-noise-ratio) and SSIM (structural similarity) between the cover and stego images. The cover image in AHDeS is the original cover image. The results are given in Table 2. It can be observed that the visual quality of the stego images for AHDeS significantly outperforms the other methods. For example, the PSNR of AHDeS, HiDDeN, SteganoGAN, CHAT-GAN, and HiNet are 45.07, 37.29, 42.61, 43.34 and 42.95, respectively.

## 5.4 Ablation Study

In this part, ablation studies are conducted from different aspects.

- Variant I. The adversary module is disabled in Variant I. Such variant degenerates into HiNet.
- Variant II. $L_{hpf}$ in the adversary module is replaced with $L_2$ loss in Variant II.
- Variant III. $L_{hpf}$ in the adversary module is replaced with the cross-entropy loss of CovNet in Variant III.
- Variant IV. The second phase of network training, i.e., alternately training the INN and optimizing the perturbations, is not adopted in the training stage.
- Variant V. CovNet is replaced with XuNet in termination condition for perturbation optimization.

The experimental results are given in Table 3 and the following conclusions can be obtained.

- Via comparing AHDeS and Variant I, it can be observed that AHDeS significantly outperforms Variant I, i.e., the original HiNet. Such results verify the effectiveness of the Dig-and-Fill paradigm.
- Via comparing AHDeS with Variant II and Variant III, it can be observed that $L_{hpf}$ is most suitable for the adversary module. Such results verify the effectiveness of the frequency compensation mechanism.
- Via comparing AHDeS and Variant IV, it can be observed that disabling the second phase of INN training would lead to obvious performance degradation. Such results verify the effectiveness of the perturbation adaptive training strategy.
- Via comparing AHDeS and Variant V, it can be observed that applying a more advanced steganalyzer in the termination condition of perturbation optimization would bring benefits.

**Table 3: Detection error rate (%), bit error rate (%), and visual quality of AHDeS and its variants.**

| Methods | CovNet | LWENet | BER | PSNR | SSIM |
|---|---|---|---|---|---|
| Variant I | 2.16 | 0.75 | **0.26** | 42.95 | 0.9893 |
| Variant II | 1.90 | 0.35 | 0.70 | 44.26 | 0.9932 |
| Variant III | 1.62 | 0.15 | 0.43 | 43.09 | 0.9898 |
| Variant IV | 2.20 | 1.20 | 0.40 | 41.09 | 0.9876 |
| Variant V | 4.99 | 2.45 | 1.27 | 44.07 | 0.9919 |
| **AHDeS** | **8.99** | **7.37** | 0.85 | **45.07** | **0.9938** |

## 5.5 Frequency Analysis

In this part, the embedding patterns of the proposed AHDeS are analyzed in the frequency domain. DWT is first applied to process the cover and stego images. And then, components on four sub-bands can be obtained, including HH, HL, LH, and LL, where H and L denote high frequency and low frequency along a specific direction, respectively. For a specific steganographic method and a specific sub-band, the difference between the components of cover and stego images can be calculated. The frequency characteristics is analyzed by visualization and statistics as follows.

From the perspective of visualization, we show the absolute difference of high frequency components and low frequency components, where the high frequency components are the summation of HH, HL, and LH sub-bands, while the low frequency components are the LL sub-band. AHDeS and HiNet is compared, and the results are given in Fig. 4. From the third and fifth columns, it can be observed that both of the AHDeS and HiNet can well preserve the low frequency components. However, from the second and fourth columns, it can be observed that HiNet leaves more obvious artifacts on high frequency components than AHDeS.

From the perspective of statistics, the values of the absolute difference on a specific sub-band are summed up. Five steganographic methods are compared, and the results averaged on 10,000 images from COCO dataset are given in Fig. 5. Comparing with HiDDeN, SteganoGAN, and CHAT-GAN, the proposed AHDeS can better

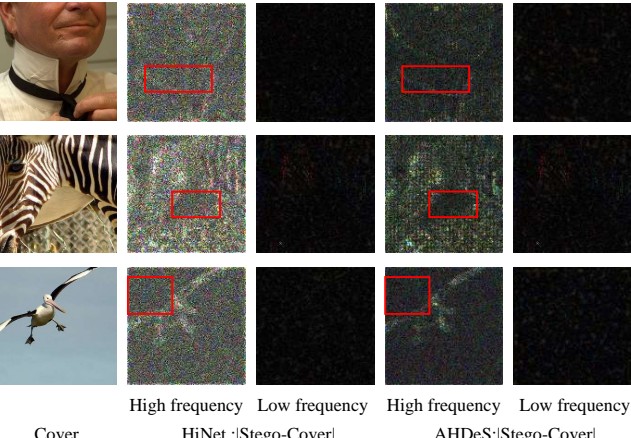

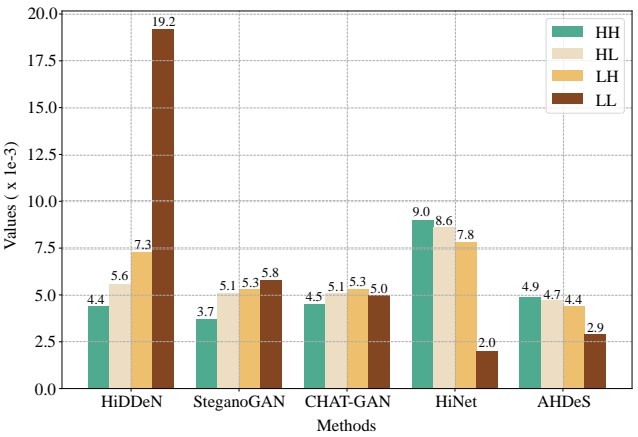

Cover

High frequency   Low frequency    High frequency   Low frequency

HiNet :|Stego-Cover|     AHDeS:|Stego-Cover|

**Figure 4: Visualization of difference between cover and stego images for high frequency and low frequency components. The difference is magnified by 20 times.**

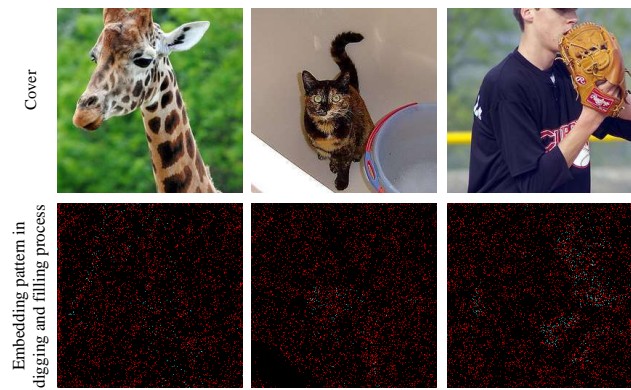

**Figure 6: Visualization of offset effect. The red and blue points represent elements in $D_{dig}$ and $D_{fil}$ having the opposite signs and same signs, respectively.**

the opposite signs and the same signs, respectively. From Fig. 6, it can be observed that the red points are far more than the blue points. Such phenomenon indicates that the modification direction in the digging and filling processes have the offset effect, and thus the stego image is closer to the original cover image. In such manner, the stego image is of better anti-steganalysis performance.

### 5.7 Robustness to JPEG compression

In this part, INN's forward calculation outputs spatial stego image, which is transformed by DCT, divided by quantization step, and processed by simulated rounding function to approximate JPEG compression. The output is decompressed as spatial image, and fed into reverse calculation for message extraction. BERs are 0.99%, 0.52% and 0.24% for quality factor of 70, 80 and 90.

### 6 Conclusions

In this paper, the steganographic method called AHDeS is proposed under the Dig-and-Fill paradigm, wherein the adversary component for countering against steganalyzers and the hiding component for information hiding can be decoupled. Extensive experiments have been conducted, and the following conclusions can be made. Firstly, AHDeS can obtain the state-of-the-art security performance and visual quality, while maintaining satisfied message extraction accuracy. Secondly, owing to the frequency compensation mechanism, AHDeS can well preserve the high frequency components without disturbing the low frequency components. Thirdly, the digging and filling processes in AHDeS have the offset effect of modification direction, which is beneficial to generating more secure stego image during information hiding. In the future, we hope to extend the Dig-and-Fill paradigm to information hiding in other multimedia carriers, such as audio and video.

### Acknowledgments

This work is supported by National Key R&D Program of China under Grant 2023YFF0905000.

**Figure 5: Statistics of components' difference between cover and stego images on four sub-bands.**

preserve the LL sub-band unchanged. Comparing with HiNet, although the effect on LL sub-band is quite similar, AHDeS can obtain much better preserving performance on high frequency sub-bands of HH, HL, and LH.

### 5.6 Offset Effect Visualization

In this part, the offset effects of the digging and filling processes in AHDeS are visualized. Note that in the proposed AHDeS, in the deployment stage, the adversary module receives the original cover image $C_{ori}$ and outputs the optimized cover image $C_{opt}$, and then the INN takes in $C_{opt}$ and secret message $M$ and outputs the stego image $S$. Specifically, the difference between $C_{opt}$ and $C_{ori}$ in the digging process is denoted as $D_{dig}$, and the difference between $S$ and $C_{opt}$ in the filling process is denoted as $D_{fil}$. The offset effect for the digging and filling processes are visualized in Fig. 6, where the red and blue points represent elements in $D_{dig}$ and $D_{fil}$ having

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
