# OpenReview forum: "Dig a Hole and Fill in Sand: Adversary and Hiding Decoupled Steganography"
_acmmm.org/ACMMM/2024/Conference — MM2024 Poster_

### Official Review · Reviewer_i3Fw · 2024-05-17

**Rating:** 4
**Confidence:** 3

**Summary:**

This paper proposed an adversary-hiding-decoupled steganography (AHDeS) scheme under the dig-and-fill paradigm. It decoupled the steganography into an optimization-based adversary module in the digging process and an INN-based hiding network in the filling process.

**Strengths:**

1. The dig-and-fill paradigm is interesting and easy to understand.
2. This paper is very well-written and easy to follow. The structure is clear and reasonable. The analyses of the dig-and-fill paradigm and the proposed method are full.
3. Compared to HiNet, the proposed digging process improved significantly the security of steganography.
4. This paper has conducted sufficient experiments to prove the effectiveness of AHDeS. The ablation and frequency analysis are also supportive.

**Limitations:**

1. (Very important) About the security of INN-based steganography. The INN has been proven as a kind of secure steganography method where the differences between stego image/audio and cover image/audio cannot be recognized, because the distribution of cover image/audio is the same as that of stego image/audio [1-4]. However, this paper was based on the assumption that the INN-based methods are not secure. Therefore, the reviewer is very eager to know which conclusion is correct, or how the AHDeS differs from the previous INN-based methods.

2. This paper mentioned in the Introduction that, “Therefore, these two components are two ends of the seesaw, and it is difficult to balance the tradeoff between message extraction accuracy and security performance by joint optimization with multiple loss functions.” (lines 121-124). To solve this problem, this paper proposed AHDeS. However, from the results in Table 1, AHDeS achieved better security but lower extraction accuracy compared to HiNet. It seems that AHDeS also suffered from the imbalance between security and extraction accuracy. Therefore, the reviewer also wants to know whether and how this paper solves the above imbalance.

3. (Important) About the steganography baselines and steganography analysis methods. The adopted methods are a little old, and the authors are expected to select recent methods (2023 or 2024) to increase the convincingness.

4. (Very important) Detection error rate (DER) in Table 1 needs to be explained in detail by the authors. To the knowledge of the reviewer, the steganography analysis model is used to judge whether the stego image contains hidden information, and output a binary classification result. AHDeS can achieve only 8.99% of DER under CovNet. Does this mean that most stego images can still be detected by steganography analysis models? From this point of view, the method proposed by the author does not seem to have its due effect.

5. (Important) This paper lacked an analysis of the relationship between payload and steganography security, and extraction accuracy.

6. About the writing: $C^{ori}$ in Eq 8 was used for the first time and should be explained here.

7. (Important) Is the proposed AHDeS robust to image compression (e.g., JPEG)?

8. The template style is wrong, and the authors are expected to use the correct template style.

[1] HiNet: Deep Image Hiding by Invertible Network

[2] Large-capacity image steganography based on invertible neural networks

[3]  Distribution-preserving steganography based on text-to-speech generative models

[4]  Hiding video in audio via reversible generative models

Note: The reviewer will change the score according to the authors' responses.

**Suitability:**

3

---

### Official Review · Reviewer_K7N5 · 2024-05-20

**Rating:** 4
**Confidence:** 4

**Summary:**

This paper proposes a novel dig-and-fill steganographic paradigm that combines the learning-based and optimization-based steganography. The authors first learn invertible steganographic network (ISN) for data embedding and extraction.  Then, based on the trained ISN (with fixed parameters), they optimize the cover images to generate undetectable stego-images. The learning and optimization processes are conducted alternately.

**Strengths:**

- The paper is well written and organized and I enjoy reading it.
- The authors provide a very clear description of the dig-and-fill paradigm, which I really appreciate.
- I appreciate the author's use of quantized images (8x3bpp) for testing.
- This paper achieves good steganography performance, especially in undetectability.

**Limitations:**

- The idea of combining the learning-based and optimization-based steganography have been explored by previous method LISO [R1]. Please detail the key differences between the proposed AHDeS and LISO.

- The results in Tables 1/2/3 indicate that the proposed AHDeS outperforms HiNet in terms of the quality/undetectability of stego-images but is inferior to HiNet in terms of BER recovery. However, there is a trade-off between the recovery BER and the quality/undetectability of the stego-images. One can improve the quality/undetectability of the stego-images by HiNet via decreasing the weight of the recovery loss in its total losses, at the cost of increasing the recovery BER. Therefore, the experimental results of the paper fail to demonstrate that the proposed method is superior to HiNet.

- How did the authors train the steganalysis network, including known CovNet and unknown XuNet/SRNet/LWENet. Please detail the dataset and training strategy.

If authors address my concerns, I will raise my score.

[R1] Chen X, Kishore V, Weinberger K Q. Learning iterative neural optimizers for image steganography[C]//The Eleventh International Conference on Learning Representations. 2022.

**Suitability:**

3

---

### Official Review · Reviewer_tHyS · 2024-05-23

**Rating:** 6
**Confidence:** 4

**Summary:**

This paper proposes an adversary-hiding-decoupled steganographic method based on the Dig-and-Fill paradigm. Specifically, the INN is first trained to implement message embedding and extraction in the digging process. And then given the trained INN, the optimization module is utilized to iteratively optimize the cover image to improve security performance without sacrificing message extraction accuracy in the filling process. The proposed method is compared with several SOTA methods from different aspects. Results show that the proposed method can significantly improve the security performance and visual quality of stego images.

**Strengths:**

1. The Dig-and-Fill paradigm is a novel instructive, which can well solve the difficulties of jointly training the adversary and hiding components in steganography. Combining the design philosophy of the Dig-and-Fill paradigm and the reversibility of INN, security performance can be enhanced without sacrificing message extraction accuracy.
2. The frequency compensation mechanism is innovative, which breaks the stereotype of merely preserving the low-frequency components unchanged in steganography.
3. Comprehensive experiments have been conducted, including performance comparison with SOTA methods and ablation studies. Results show that the proposed method can obtain SOTA performance.

**Limitations:**

The description of how to train the steganalyzer for termination conditions is missing. Provide detailed information on such training process.

-Typo. In Figure 1 (b), Diresctly->Directly.
-What images are utilized to pre-train the steganalyzer (CovNet) in the proposed method?
-How do different settings of confidence threshold and iteration number in termination condition affect the steganographic performance?

**Suitability:**

3

---

### Official Review · Reviewer_MQY3 · 2024-05-24

**Rating:** 1
**Confidence:** 4

**Summary:**

This paper proposes a steganographic method under the Dig-and-Fill paradigm, which decouples the adversary component and the hiding component. Specifically, the INN is first trained to implement the hiding task, and then the cover image is optimized to enhance the security. Experiments show that the proposed method outperform previous methods.

However, the structure of the paper is somewhat repetitive. The details of the methodology are not clearly described. Additional information and clarification are necessary to fully understand the proposed approach.

**Strengths:**

1. The topic of this paper is interesting,

2. The experimental results are compelling.

**Limitations:**

1. Line 148 and Line 151: There are two filling processes.
> the INN is first trained to implement information hiding in the **filling** process. Afterwards, given the well-trained and fixed INN, the cover  image is iteratively optimized for enhancing the security performance against steganalyzers in the **filling** process.

2. The description of the digging process and the filling process is confusing.

   In Line 133-140 and Line 342-358, the Dig-and-Fill paradigm is described as follows:
> the steganographer can first optimize the cover image in the digging process and then embed secret messages into the optimized cover image in the filling process. If the steganographer is fully aware of the hiding pattern in the filling process, then the steganographer can optimize the cover image in advance in the digging process, so that the optimized cover image embedded with secret messages can obtain satisfied security performance.

   In Line 147-151 and Line 364-370, the Dig-and-Fill paradigm is described as follows:
> In AHDeS, the INN-based hiding network is first trained to implement message embedding in the filling process, and then based on the well-trained INN, the optimization-based adversary module is utilized to optimize the cover image for enhancing security performance in the digging process.

   There is inconsistency in the description of the sequence of digging and filling. Morever, the Dig-and-Fill paradigm in Figure 2 is again a two-way process. So the description of the digging process and the filling process is rather confusing.

3. The content in Section 3 is entirely repetitive of the Introduction. Therefore, Section 3 should be removed to avoid redundancy.

4. According to the description of the Dig-and-Fill paradigm in Introduction section, the digging and filling processes are closely related, requiring knowledge of the filling process details to prepare in advance during the digging process. However, Section 4 describes the INN-based hiding process and adversarial example-based optimization processes as completely independent. The overall process involves adding adversarial noise to the cover to mislead the steganalyzer and then embedding the message into the optimized cover. This adversarial example-based optimization seems more like a preprocessing step rather than a digging process related to the filling process.

5. If the adversarial noise is added to the cover images before embedding the message, will the latter process disrupt the already added adversarial noise, reducing the possibility of successfully misleading the steganalyzer?

6. Is there only one steganalyzer used during adversarial training? If so, will the adversarial noise only defend against this specific steganalyzer, thus limiting steganographic security? The authors should conduct experiments to verify this, especially using traditional steganalyzers based on handcrafted features, to test for this steganalyzer mismatch.

7. How is the secret message input into the INN? Is the binary bitstream converted into a two-dimensional format similar to an image? What is the steganographic capacity?

8. In Table 1, some steganographic methods hide an image within another image, whereas the proposed method hides a bitstream within an image. The hiding capacities of these two types of methods are different, making the comparison unfair.

**Suitability:**

2

---

### Meta-Review · Area_Chair_3Ac6 · 2024-06-30

**Recommendation:** Accept (Poster)
**Confidence:** 3

**Metareview:**

The paper presents a novel approach to modification-based steganography. While the research has several strengths, some areas require further clarification and experimentation to enhance the study's comprehensiveness. Despite these limitations, the paper's contributions and potential impact on the field merit its acceptance.

Limitations:

The paper should clarify whether the order of digging and filling processes depends on the deployment stage or the training stage. This is crucial for understanding the methodology and its application in different scenarios.
It is unclear whether the adversarial noise is added to the cover images before embedding the message. Clarifying this aspect would help in replicating the experiments and understanding the impact of adversarial noise on the method's performance.
Hiding an image into an image and hiding a bitstream into an image are fundamentally different tasks. The paper should address this discrepancy to ensure a fair and accurate comparison of experimental results.
The paper focuses on modification-based steganography but should also include baselines from both modification-based and generative steganography methods, especially for undetectability comparisons. This would provide a more comprehensive evaluation of the proposed method.
Although the method improves undetectability compared to baselines, the Detection Error Rate (DER) is still too low, indicating potential security issues. The paper should discuss this limitation and explore ways to further enhance the method's security.
The authors are expected to provide experimental results analyzing the relationship between payload and security, as well as extraction accuracy. The current text descriptions are commonly known and not convincing. Detailed experimental data would strengthen the paper's claims and provide more robust evidence of the method's effectiveness.

The paper makes significant contributions to the field of steganography by introducing an innovative method that aims to improve undetectability and security. While there are several areas requiring further clarification and more comprehensive experimental analysis, the overall impact and novelty of the research justify its acceptance. Addressing the identified limitations in future work will further strengthen the study and its contributions to the field.